# MS$^2$-Transformer: An End-to-End Model for MS/MS-assisted Molecule Identification

## Abstract

Mass spectrometry (MS) acts as an important technique for measuring the mass-to-charge ratios of ions and identifying the chemical structures of unknown metabolites. Practically, tandem mass spectrometry (MS/MS), which couples multiple standard MS in series and outputs fine-grained spectrum with fragmental information, has been popularly used. Manually interpreting the MS/MS spectrum into the molecules (i.e., the simplified molecular-input line-entry system, SMILES) is often costly and cumbersome, mainly due to the synthesis and labeling of isotopes and the requirement of expert knowledge. In this work, we regard molecule identification as a spectrum-to-sequence conversion problem and propose an end-to-end model, called MS$^2$-Transformer, to address this task. The chemical knowledge, defined through a fragmentation tree from the MS/MS spectrum, is incorporated into MS$^2$-Transformer. Our method achieves state-of-the-art results on two widely used benchmarks in molecule identification. To our best knowledge, MS$^2$-Transformer is the first machine learning model that can accurately identify the structures (e.g., molecular graph) from experimental MS/MS rather than chemical formula/categories only (e.g., $C_6H_{12}O_6$/organic compound), demonstrating the great application potential in biomedical studies.[1]

## 1 Introduction

Mass spectrometry (MS)-based metabolomics plays a critical role in life science (Klünemann et al., 2021; Banh et al., 2021), which assists in discovering molecular biomarkers/drugs for diagnosis/therapy. MS measures the mass-to-charge (m/z) ratios of adduct ions and outputs a mass spectrum where intensity denotes the ion abundance at the corresponding m/z. In real-world applications, tandem mass spectrometry (MS/MS) is more used, where two standard MS are sequentially coupled for a more fine-grained spectrum than using one MS only (denoted as MS1). In MS/MS, the second MS (denoted as MS2) records the intensities of the fragments which are broken down from the adduct ions in MS1. As a result, MS/MS affords the capability of separating and identifying ions with close m/z-ratios in MS1 based on their fragments in MS2. An example is shown in Figure 1(a), where multiple peaks exist in different m/z ratios. In MS/MS spectrum, the correlations of peaks determine the strengths of chemical bonds, and the locations of peaks indicate the masses of potential functional groups.

To identify the molecules based on MS/MS, conventionally, there are two types of approaches. The first one is accurate mass analysis, in which an experimentally measured mass determines the elemental composition by mass matching. However, such an approach has high false-positive rates, as shown in (Kind & Fiehn, 2006; Shen et al., 2019). The other approach is database searching (DS), in which a matching score between the experimentally obtained spectrum and the database made up of standard spectra is calculated. However, DS does not bring significant improvement in molecule identification due to the limitation of the database. For example, about 90% of known metabolites don't have reference tandem mass spectra in METLIN (https://metlin.scripps.edu/) and HMDB (http://www.hmdb.ca/) databases (Shen et al., 2019; Vinaixa et al., 2016). To overcome this difficulty, researchers turn to more careful chemical designs. For example, people use $^{13}$C-labelled isotopes to determine the number of carbons (Tsugawa et al., 2019), but the cost is too high (about $2000 per gram).

---

[1]Code is available at https://github.com/bmebmebme/ms2transformer

Correspondingly, identifying metabolites with machine learning methods is a new trend and progress has been achieved in metabolite classification by neural networks recently (Dührkop et al., 2021). However, rough classification of chemical categories, like lipids or benzenoids, still cannot solve the acute problem of identifying the specific structure for profound medical research (e.g., the accurate structure recognition in antibody-antigen interaction), leaving great space for MS/MS identification.

In this work, we regard the task as a spectrum-to-sequence task, where the input is the MS/MS spectrum, and the output is the molecule structure, which is represented by the simplified molecular-input line-entry system (SMILES). Considering that chemists often leverage fragmentation tree (Figure 1b) for molecule identification, which describes the relations of the peaks in the spectrum, we incorporate it into our model. In fragmentation tree, the nodes represent the peaks, and the edges represent the fragmentation reactions between peaks. The detailed calculation process is described in Figure 1b, mainly including three steps: initialization of fragmentation graph, scoring of fragmentation graph, and ranking of fragmentation trees. A fragmentation tree can be obtained by the SIRIUS chemical tool (Dührkop et al., 2019).

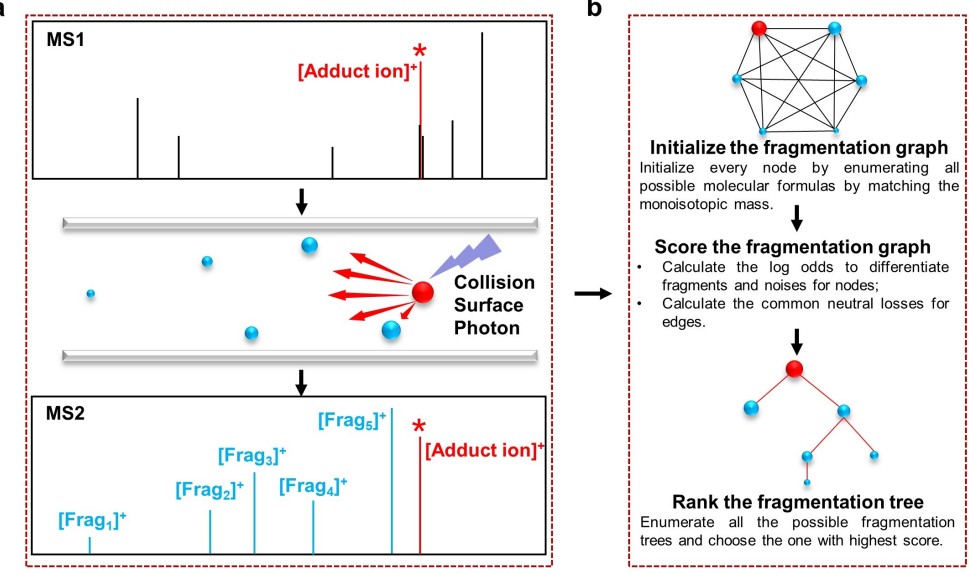

Figure 1: Workflow of MS/MS and fragmentation tree. **(a)** Illustration for adduct ions and fragmentation of metabolites. Metabolites are firstly adducted by selected cations to form [Adduct ion]$^+$ in MS1. Then, [Adduct ion]$^+$ is fragmented to form fragments in an inert gas atmosphere, and the resulted fragments are recorded by MS2. **(b)** Calculation process of fragmentation tree. Firstly, initialization of fragmentation graph is performed by enumerating all possible molecular formulas for every node by matching the monoisotopic mass. Secondly, scoring is calculated separately for nodes and edges. For nodes, calculate the mass difference between the measured peak and the molecular formula, then compute log odds to differentiate the fragment and the background noise. The principle behind this is that mass differences for fragments are assumed to be Normal distributed, while exponentially distributed for noises. For edges, calculate the common neutral losses, which are typically described as the loss of small molecules (e.g., water and ammonia), not ions. Thirdly, ranking of fragmentation trees is performed by choosing the one with highest score among all the possible fragmentation trees.

We propose an end-to-end model, MS$^2$-Transformer, for MS/MS assisted molecule identification. MS$^2$-Transformer follows the encoder-decoder framework, and it takes both the spectrum and the fragmentation tree as input. Our model consists of three modules: (1) *Peak embedding module*, where we map the (m/z ratio, intensity) pairs to high-dimension vectors. The embeddings are used for each peak in the spectrum and each node in the tree. (2) *Fragmentation aggregation module*, where the encoder of MS$^2$-Transformer consists of stacks of blocks. Each block is made up of a self-attention layer (used to process the spectrum), a graph encoding layer (used to process the fragmentation tree), and a feed-forward layer (used to fuse the features extracted by the above two

layers). In this way, the global information in the spectrum and the local information in the tree are fully leveraged. (3) *SMILES reconstruction module*, where the decoder is the standard Transformer decoder, which outputs the SMILES based on the output of the encoder. We conduct experiments on widely used datasets for MS/MS, MassBank and CASMI, and have achieved state-of-the-art results.

The main contributions of our work are as follows:

- We propose MS$^2$-Transformer, a machine learning-based model to identify the molecule with structural information based on Fragmentation Aggregation (FA) module and Transformer. Experiments on benchmark datasets validate that MS$^2$-Transformer achieves state-of-the-art performance.

- We interpret the representation of MS/MS data as information aggregation on chemical bonds globally and locally. The global aggregation on chemical bonds is performed among all peaks, referred as a fully-connected graph. The local aggregation on bonds is performed among partial peaks, which are estimated by the precursor-product relationship via fragmentation tree, referred as a locally-connected graph.

- We introduce self-attention mechanism and message passing neural network in the aggregation on fully- and locally connected graphs for learning the representation of bonds' strengths from MS/MS data.

## 2 PROBLEM FORMULATION AND NOTATIONS

Let $\mathcal{X}$ and $\mathcal{Y}$ denote source domain and target domain, which are collections of MS/MS spectrum and molecules, respectively. We assume that the molecules are represented by the canonical SMILES, which provides a unique representation string for one molecule. Our task is to learn a mapping $f : \mathcal{X} \mapsto \mathcal{Y}$, that can identify the molecule from the input spectrum. We focus on the supervised setting, where the dataset consists of $M$ spectrum-molecule pairs $\mathcal{D} = \{(x_i, y_i)\}_{i=1}^{M}$, $x_i \in \mathcal{X}, y_i \in \mathcal{Y}$.

Each $x_i$ can be further decomposed into $x_i = \{(m_j, I_{m_j})\}_{j=1}^{N_i}$, where $m_j$ denotes the m/z ratio, $I_{m_j}$ represents the corresponding ion intensity, and $N_i$ denotes the number of peaks in the spectrum. Note that the m/z ratio is discrete, while the intensity is continuous.

Distinct yet limited patterns can be summarized from MS/MS spectra, providing assistance for the chemists to infer the molecular candidates from bio-fluids. The patterns carrying prior knowledge can be divided into monoisotopic mass and fragmentation patterns.

Monoisotopic mass calculates the molecular mass by summing up the atoms' masses with their most abundant naturally occurring stable isotopes, and it can provide assistance in enumerating all the potential chemical formulas. For example, as shown in Figure 2a, the recorded m/z ratio of 132.1030 Da with a negligible shift of 1.0949 mDa from its monoisotopic mass, could help infer the potential candidate with a chemical formula of $C_6H_{13}NO_2$.

The fragmentation pattern is derived from the correlations among peaks, and it can help find potential functional groups that constitute the molecule. For example, as illustrated in Figure 2a, $\Delta$m/z of 18 Da may denote the dehydration process of losing an $H_2O$, indicating that there exists a hydroxyl in the target molecule's chemical structure.

Thus, based on the monoisotopic masses and the fragmentation patterns, researchers can enumerate all the possible chemical formulas and functional groups, making it feasible to calculate the hypothetical fragmentation tree by maximum likelihood from the MS/MS spectrum, as shown in Figure 2b. The nodes refer to the peaks in MS/MS, and edges refer to the fragmentation correlations between peaks (Böcker & Dührkop, 2016; Dührkop et al., 2019).

## 3 OUR MODEL

We introduce our proposed model in this section. Our model consists of three modules, the peak embedding module, the fragmentation aggregation module, and the SMILES reconstruction module. We also introduce a training technique used in our method. The overall framework is in Figure 3.

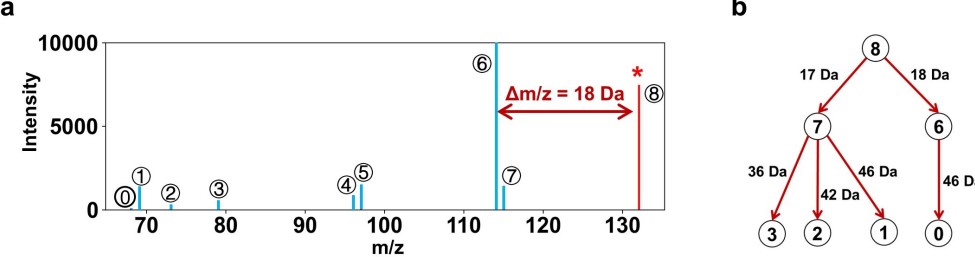

Figure 2: Demonstration of one experimental MS/MS. **(a)** Typical experimental MS/MS of aminocaproic acid, in which the monoisotopic mass is marked with a red star and the fragmentation pattern of dehydration is indicated by $\Delta$m/z of 18 Da. **(b)** The fragmentation tree of aminocaproic acid calculated from **(a)**.

### 3.1 PEAK EMBEDDING MODULE

We use an m/z embedding $E_{m/z} \in \mathbb{R}^{v \times d}$ and an intensity embedding $E_{\text{int}} \in \mathbb{R}^{v \times d}$ to encode the input. $E_{m/z}$ is retrieved from the embedding $E_M \in \mathbb{R}^{V \times d}$, $V$ is the number of distinct m/z ratios in the dataset, and $v$ is the number of m/z ratios in one spectrum. Mathematically, the input m/z vector $M$ (binned with a size of 1 Da) and the intensity vector $I$ is processed as follows:

$$h^0 = W_o \text{concat}(E_{int}, E_{m/z}); E_{int} = W_{int}I; E_{m/z} = E_M(M), \tag{1}$$

where $\text{concat}$ means that the inputs are concatenated along the last dimension, $W_o \in \mathbb{R}^{d \times d}$ and $W_{int} \in \mathbb{R}^{1 \times d}$ are the parameters to be learned. The ion intensity $I_i$ is firstly projected into a $d$-dimension vector, then concatenated with the embedding of its corresponding m/z. In this way, the input m/z vector $M$ and intensity vector $I$ are encoded as $h^0 = (h_1^0, h_2^0, \cdots, h_v^0)$.

### 3.2 FRAGMENTATION AGGREGATION MODULE

Given an MS/MS spectrum $x_i$, the corresponding fragmentation tree $T_i$ can be obtained from $x_i$ with the external chemical tools (Dührkop et al., 2019). Note that the number of nodes in $T_i$ is the same as the number of element in $x_i$, i.e., $N_i$. We regard $T_i$ as an undirected graph, and use $\mathcal{E}_i$ to denote the edges in $T_i$. For each node $j$ in $T_i$, denote the neighbors as $\mathcal{N}(j) = \{k | (k, j) \in \mathcal{E}_i\}$. The fragmentation aggregation module consists of $L$ blocks, where each block consists of a local aggregation module (i.e., a graph layer), a global aggregation module (i.e., a self-attention layer) and a module to merge the two representations.

*Local aggregation*: We performed the local information aggregation on the fragmentation tree $T_i$ with MPNN (Gilmer et al., 2017). Let $h_i^l$ denote the features of node $i$ output by the $l$-th block. We set $\{h_i^0\}$ as the embeddings output by the peak embedding module. Mathematically, the $l$-th block of local aggregation works as follows:

$$h_{\text{local},i}^l = W_s h_i^{l-1} + \frac{1}{|\mathcal{N}(i)|} \sum_{j \in \mathcal{N}(i)} W_n h_j^{l-1}, \tag{2}$$

where $W_s$ and $W_n$ are network parameters. That is, we aggregate local features on the tree using GraphSage (Hamilton et al., 2018), and obtain $h_{\text{local},i}^l$.

*Global aggregation*: In parallel, we use a self-attention layer to globally process all $h_i^{l-1}$'s. A self-attention layer is defined as follows:

$$h_{\text{global},i}^l = \sum_{j=1}^{N} \alpha_j W_v h_j^{l-1}; \; \alpha_j = \frac{\exp(h_i^{l-1} W_q h_j^{l-1})}{\sum_{k=1}^{N} \exp(h_i^{l-1} W_q h_k^{l-1})}, \tag{3}$$

where $W_v$ and $W_q$ are parameters to be learned.

After that, we utilize both local and global features of the $l$-th block and obtain

$$h_i^{\text{all}} = h^{l-1} + \text{LN}(h_{\text{local},i}^l + h_{\text{global},i}^l), \tag{4}$$

where LN denotes layer normalization (Ba et al., 2016).

After that, $h_i^{\text{all}}$ is fed into a feed-forward layer and $h_i^l$ is obtained. Mathematically,

$$h_i^l = W_2 \text{ReLU}(W_1 h_{\text{all},i}^l + b_1) + b_2, \tag{5}$$

where $W_1$, $W_2$, $b_1$, $b_2$ are to be learned.

We can eventually obtain $H^L = (h_1^L, h_2^L, \cdots, h_N^L)$ from the last block of the encoder.

## 3.3 SMILES RECONSTRUCTION MODULE

We use the standard Transformer decoder (Vaswani et al., 2017) to generate the molecule following the teacher forcing training strategy with the ground truth from a prior time step as input. The decoder consists of a self-attention layer, encoder-decoder attention layer, and feed-forward layer. The $H^L$ will be used in the encoder-decoder attention, and the SMILES will be generated token by token.

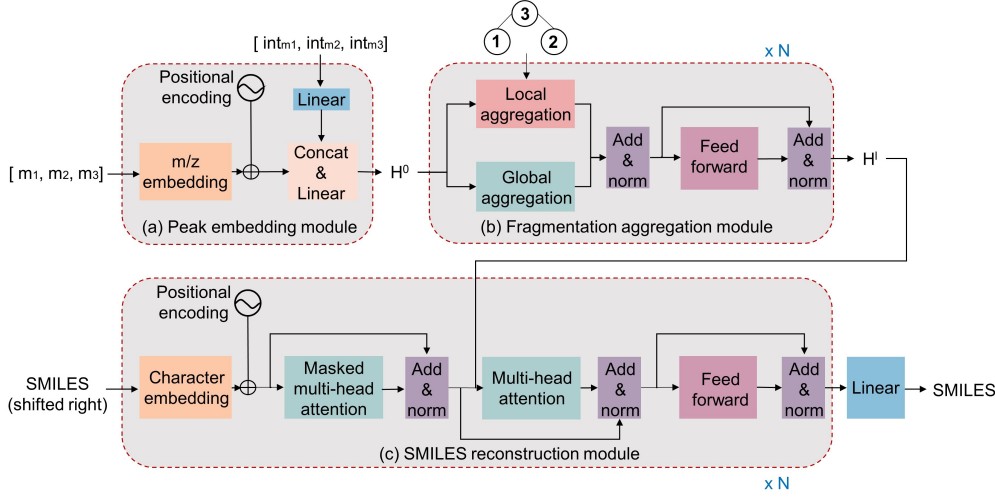

Figure 3: The architecture of MS$^2$-Transformer. In peak embedding module, the m/z's are processed by an embedding layer with positional encoding, and the intensities are first projected into the same dimension as the m/z's embedding, then concatenated with m/z embedding for the construction of peak embedding by a Linear layer. In the fragmentation aggregation module, the peak embedding was updated by local aggregation on the fragmentation tree, and global aggregation on the fully connected graph, where the nodes denote the peaks in the spectrum and the edges denote the fragmentation correlations. In the SMILES reconstruction module, the learned peak embedding was used for SMILES reconstruction by Transformer.

## 3.4 DROPPING MODULE

To relieve the overfitting effects, inspired by (Larsson et al., 2017; Zhu et al., 2020), we design an aggregation switch to control the choice of local and global fragmentation aggregation. Mathematically, during training, Eqn.(4) is refined as followed:

$$h_i^{\text{all}} = h_i^{l-1} + \begin{cases} \text{LN}(h_{\text{local},i}^l), & 0 \le p < p_{\text{local}}; \\ \text{LN}(h_{\text{global},i}^l), & p_{\text{local}} \le p < p_{\text{local}} + p_{\text{global}}; \\ \text{LN}(h_{\text{local},i}^l + h_{\text{global},i}^l), & p_{\text{local}} + p_{\text{global}} \le p \le 1; \end{cases} \tag{6}$$

where $p$ is uniformly sampled from $[0,1]$, $h_{\text{local},i}^l$ and $h_{\text{global},i}^l$ are defined in Eqn.(2) and Eqn.(3) respectively. With probability $p_{\text{local}}$, we only use the local aggregation; with probability $p_{\text{global}}$, we

Table 1: Summary of benchmark datasets.

| Dataset | #Training pairs | #Testing pairs | #Molecules |
|---------|-----------------|----------------|------------|
| MassBank | 15,784 | 3,945 | 3,675 |
| CASMI | 380 | 94 | 393 |

use the global aggregation; otherwise, we use both of them. At inference time, both of the two branches are used as Eqn.(4).

In this work, we focus on spectrum-to-SMILES. The proposed fragmentation aggregation method is general and can be easily applied to other MS/MS-based applications like metabolite classification.

# 4 EXPERIMENTS

## 4.1 DATASETS AND BASELINE MODELS

We performed the experiments on the benchmark datasets of electrospray ionization (ESI) MS/MS spectra of the positive ionization mode from **MassBank** (Horai et al., 2010) and the MS/MS spectra of the Critical Assessment of Small Molecule Identification (**CASMI**) Contest (Nikolic et al., 2017; Schymanski et al., 2017). We split the dataset randomly by spectrum, because the cosine similarity score (CSC) for intra-molecule was low, with a mean CSC of 0.38.

The MassBank dataset consists of 19,729 spectrum-SMILES pairs and 3,675 distinct molecules in Figure 4. It covers a wide range of chemical compounds, including benzenoids, alkaloids, organic acids, and lipids, etc. The length of SMILES varies from 5 to 100, the number of edges in the fragmentation trees fluctuates from 1 to 59, and the number of peaks in MS/MS covers the range from 2 to 189. The CASMI dataset consists of 474 spectrum-SMILES pairs and 393 distinct molecules in Figure 4. The length of SMILES varies from 5 to 93. The number of edges in the fragmentation trees fluctuates from 1 to 58, the number of peaks in MS/MS covers the range from 2 to 185, and the precursor mass of both MassBank and CASMI is below 500 Da.

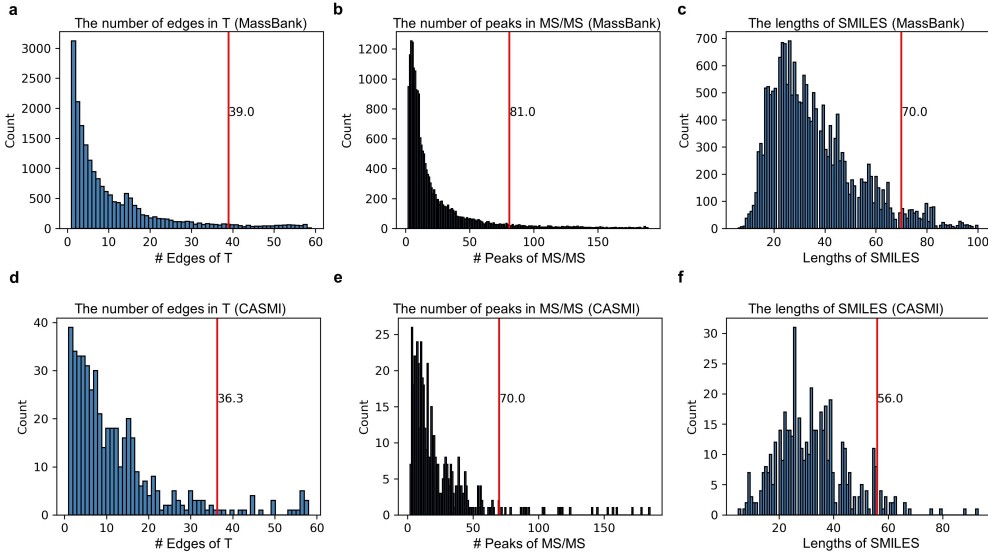

Figure 4: The data statistics of the benchmark datasets. The distribution for the number of (**a**) edges in $T$ and (**b**) peaks in MS/MS, and (**c**) the lengths of SMILES in MassBank dataset. The distribution for the number of (**d**) edges in $T$ and (**e**) peaks in MS/MS, and (**f**) the lengths of SMILES in CASMI dataset. The red line indicated the 95% percentile.

The reconstruction quality of molecule identification was tested based on validity of valency constraints (denoted as V), reconstruction accuracy with valid valency (denoted as R), top-1 and -5 reconstruction accuracy (denoted as Top1 and Top5), Formula 1 score (denoted as F1 score), mean rank of the correct prediction (denoted as mean rank), mean relative ranking position (denoted as mean RRP). F1 score is similar to the scheme in F1 racing based on the rank of the correct prediction. R, V, and top1/top5 accuracy are widely used in molecule reconstruction problem (Shi et al., 2020; Jin et al., 2019), while F1 score, mean rank, and mean RRP are widely used in the performance evaluation of CASMI (`http://casmi-contest.org/2017/index.shtml`). F1 score, mean rank, and mean RRP are calculated with the top-10 candidates. Above these measurements, state-of-art models perform better with larger R, V, top1/top5, F1 score, and mean RRP, and smaller mean rank. The top-k candidates were generated by beam-search.

We chose the following models as the baselines: **MetFrag** (Ruttkies et al., 2016) represented the typical conventional database-searching model for MS/MS-assisted molecule identificaiton.MetFrag first generated in silico fragmentation spectra of molecules from different databases, then matched against m/z ratios. MetFrag was downloaded from `https://ipb-halle.github.io/ MetFrag/projects/metfragcl/`(version 2.4.5). The searching databases for MetFrag included PubChem and KEGG and the identification was performed with a mass acuuracy of 0.1 Da. **RNN** first encoded the spectrum into hidden representations, then generated the SMILES in a seq2seq architecture with Luong attention mechanism and LSTM operators. We implemented the model using the code shared at `https://github.com/ywk991112/pytorch-chatbot`. **Transformer** represented the classic NLP model, and it first encoded the hidden representations with self-attention mechanism then generated the SMILES with self- and cross-attention mechanisms. We implemented the model using the code shared at `https://github.com/ jadore801120/attention-is-all-you-need-pytorch`.

### 4.2 TRAINING DETAILS

As we view MS/MS-assisted molecule identification as the translation from tandem mass spectrum to SMILES string, which describes the structure of chemical molecules using short ASCII strings, the loss function is defined by multi-class cross entropy loss as follows:

$$\mathcal{L} = -\sum_{k=1}^{K} y_{hot}^k \log p_k, \tag{7}$$

where $K$ represents the number of label classes, $p_k$ represents the predicted probabilities of ASCII characters, and $y_{hot}$ represents the original one-hot encoding on ASCII characters.

In order to improve generalization and model calibration, we estimate the network's parameters with label smoothing by minimizing the following loss function:

$$y_{ls} = (1 - \epsilon) * y_{hot} + \epsilon u(k); \mathcal{L} = -\sum_{k=1}^{K} y_{ls}^k \log p_k, \tag{8}$$

where $u(k)$ represents the prior distribution of ASCII characters.

During training, we used the Adam optimizer (Kingma & Ba, 2017) with varied learning rate (Vaswani et al., 2017) as follows:

$$lr = d^{-0.5} * min(step\_num^{-0.5}, step\_num * warmup\_steps^{-1.5}), \tag{9}$$

where the learning rate was firstly increased during the warm_up steps and then decreased. We trained our models on one machine with four NVIDIA GeForce RTX 2070 Super GPUs with a batch size of 64 and a training epoch of 500.

### 4.3 EVALUATION RESULTS ON MOLECULE IDENTIFICATION

**MassBank**: Table 2 shows the performance of the models in the MassBank dataset. Compared to the conventional database-searching model, all deep learning models have superior performances on all metrics. $MS^2$-Transformer has achieved competitive performances on the seven metrics. As a model introducing the prior chemical knowledge, $MS^2$-Transformer has improved the top-1 accuracy and

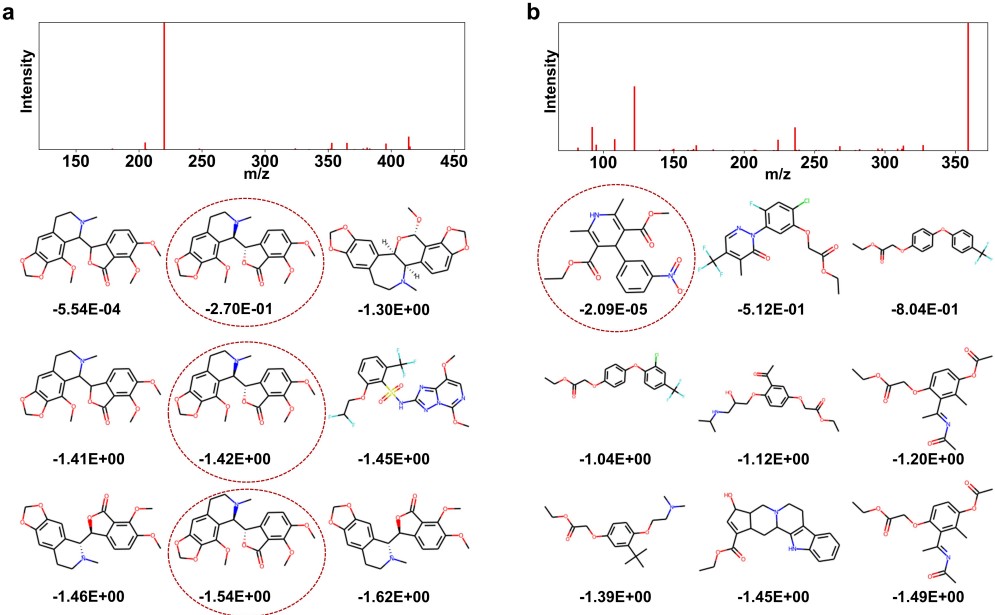

Figure 5: The demonstration of generated SMILES in **(a)** MassBank and **(b)** CASMI dataset. The above spectrum denotes the MS/MS spectrum as the input of MS$^2$-Transformer, the below molecular graphs denote the top-9 candidates generated with beam search, the number below molecular graph indicates the searching score, and the circled molecule denotes the accurate prediction of true molecule.

Table 2: Comparison of different models' performance on MassBank dataset.

| Model | V% | R% | Top1% | Top5% | F1 score | Mean rank | Mean RRP |
|---|---|---|---|---|---|---|---|
| MetFrag-PubChem | - | - | 0.10 | 0.48 | 328 | 9.96 | 0.04 |
| MetFrag-KEGG | - | - | 1.14 | 1.55 | 1,405 | 9.85 | 0.02 |
| RNN | 77.4 | 59.0 | 45.7 | 47.0 | 45,953 | 5.32 | 0.47 |
| Transformer | **97.7** | 58.0 | 56.7 | 70.1 | 64,980 | 3.11 | 0.69 |
| MS$^2$-Transformer | 97.1 | **63.2** | **61.4** | **73.7** | **68,736** | **2.77** | **0.72** |

reconstruction accuracy by 5%, compared with Transformer. Also, MS$^2$-Transformer improved the performances evaluated by F1 score, mean rank, and mean RRP.

**CASMI**: Table 3 shows the performance of the models in the CASMI dataset. Compared to the conventional database-searching model, all deep learning models have superior performances on all metrics. Due to insufficient data pairs in the CASMI dataset, the top-5 identification accuracy only achieves 52.1%. MS$^2$-Transformer still shows superior performances to Transformer, with an improvement of 8%/163 in reconstruction accuracy/F1 score, respectively. Importantly, there were 73 distinct molecules which hadn't been seen in the training dataset, and the top-1/5 accuracy achieved 40.79%/52.63% for these independent molecules, respectively.

By the way, MS$^2$-Transformer has obtained the above performances in both MassBank and CASMI only with additional 0.026 M parameters in a light-cost way, compared to the conventional transformer.

## 5 CONCLUSIONS AND FUTURE WORK

In this paper, we propose the MS$^2$-Transformer for reconstructing the right chemical structures from MS/MS spectrum based on the aggregation of prior chemical knowledge. Our method identifies the

Table 3: Comparison of different models' performance on CASMI dataset.

| Model | V% | R% | Top1% | Top5% | F1 score | Mean rank | Mean RRP |
|-------|-----|------|-------|-------|----------|-----------|----------|
| MetFrag-PubChem | - | - | 0.0 | 3.2 | 61 | 9.65 | 0.04 |
| MetFrag-KEGG | - | - | 3.19 | 5.32 | 111 | 9.49 | 0.05 |
| RNN | 78.7 | 40.5 | 31.9 | 31.9 | 750 | 6.81 | 0.32 |
| Transformer | **95.7** | 34.4 | 33.0 | 46.8 | 971 | 5.57 | 0.44 |
| $MS^2$-Transformer | **95.7** | **42.2** | **40.4** | **52.1** | **1,134** | **4.91** | **0.51** |

right chemical structures of molecules and exploits the fragmentation inferences for potential fragment candidates. Experimental results showed that $MS^2$-Transformer outperformed Transformer in the benchmark datasets of molecule identification with an improvement of 5%-7% in top-1 accuracy.

For further improvement, we plan to extend the $MS^2$-Transformer to problems involving multiple ionization modes (i.e., both positive and negative ionization modes) to investigate the fragmentation aggregation mechanisms in different experimental settings. We also plan to extend our model to encoding the MS/MS spectrum of mixtures, providing help in the clinical research of molecule identification from complex bio-fluids, not only from the pure substances.

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
