# OpenReview forum: "MS$^2$-Transformer: An End-to-End Model for MS/MS-assisted Molecule Identification"
_ICLR.cc/2022/Conference — ICLR 2022 Submitted_

### Official Review · Reviewer_PCDA · 2021-10-21

**Correctness:** 3
**Technical Novelty And Significance:** 2
**Empirical Novelty And Significance:** 2
**Recommendation:** 5
**Confidence:** 5

**Main Review:**

This is an interesting paper that exploits sequence to sequence machine learning models to solve mass spectrometry problems.
The strength of this paper is that they incorporated extra chemical knowledge of the spectrum via its fragmentation tree to enhance the input feature.
The weakness of this paper mainly comes from its experimental settings:
(1) The paper doesn't explain how many different molecules/compounds are in the MassBank dataset and the CASMI dataset.
(2) How are your training, validation, and testing sets split? The MassBank dataset contains spectrum records of the same molecule but at different experimental conditions (e.g. different collision energy). On average there are around 10 records per compound. Did you split your training, validation, and testing sets randomly or by compounds? This makes a huge difference in terms of performance. The former would be an easy task due to the high overlapping of known compounds in the training set, which makes it more like an overfitting problem. The latter should be the right way to test the model's performance.
(3) How is your reconstruction loss defined?
(4) What is your label in equation (7)?
(5) The same molecular structure could have many different SMILES representations. How does your model address this issue (non-deterministic mapping)?
(6) They author doesn't provide a code to reproduce their experiments.

**Summary Of The Paper:**

This paper proposed a transformer model for molecule structure identification using the experimental mass spectra.
This model takes the experimental mass spectrum and the corresponding fragmentation tree structure as the input for the transformer to predict the SMILES of the molecule in an end-to-end manner.
In their experiments, they show their model improves a vanilla transformer by 5% top-1 accuracy.

**Summary Of The Review:**

This paper proposed a customized transformer to solve the molecular structure identification problem via MS/MS spectrum. The application itself is interesting but the authors need to justify their performance with a clear experimental setting and a better illustration of the evaluation process.
With the current paper, I this it is marginally below the acceptance threshold. If you address my questions properly, I may adjust my score.

---

> ### Author Response · Authors · 2021-11-23
> **Response to Reviewer PCDA**
>
> Comment: (1) The paper doesn't explain how many different molecules/compounds are in the MassBank dataset and the CASMI dataset.
>
> Response: We thank the reviewer for the helpful comment. The number of distinct molecules is 3,675 and 393 for MassBank and CASMI dataset, respectively. We add the information of the statistics for molecules/compounds in Table 1.
>
> Comment: (2) How are your training, validation, and testing sets split? The MassBank dataset contains spectrum records of the same molecule but at different experimental conditions (e.g. different collision energy). On average there are around 10 records per compound. Did you split your training, validation, and testing sets randomly or by compounds? This makes a huge difference in terms of performance. The former would be an easy task due to the high overlapping of known compounds in the training set, which makes it more like an overfitting problem. The latter should be the right way to test the model's performance.
>
> Response: We thank the reviewer for the thoughtful comment. We split the dataset randomly by spectrum due to the low cosine similarity score (CSC) of spectra for intra-molecule (mean CSC of 0.38).
>
> Besides, the number of independent molecules which haven’t been seen in the training dataset was 73 for CASMI. For these molecules haven’t been seen in training dataset, our model achieves a top-1 accuracy of 42.79% and a top-5 accuracy of 52.63%, which is close to the performance of the whole test dataset.
>
> Comment: (3) How is your reconstruction loss defined?
>
> Response: We thank the reviewer for the helpful comment. The loss function is defined by cross entropy loss. Specifically, we view MS/MS-assisted molecule identification as the translation from tandem mass spectrum to SMILES representation, which describes the structure of chemical molecules using short ASCII strings. Thus, the prediction of SMILES leads to a multi-class classification problem and the loss is defined as a multi-class cross-entropy loss as follows:
> \begin{equation}
> \mathcal{L}=-\sum_{k=1}^{K} y_{hot}^{k} \log p_{k},
> \end{equation}
>
> Comment: (4) What is your label in equation (7)?
>
> Response: We thank the reviewer for the thoughtful comment. The label in equation (7) represents the class label of ASCII characters, which construct the SMILES string.
>
> Comment: (5) The same molecular structure could have many different SMILES representations. How does your model address this issue (non-deterministic mapping)?
>
> Response: We thank the reviewer for the helpful comment. We trained the model with tandem mass spectrum and the canonical SMILES representation. Canonical SMILES is unique SMILES with isomeric information, which encodes isotopes, charges, radicals, stereocenters, stereogroups, cis-trans bonds, and aromaticity into SMILES in a canonical form. One molecule only has one canonical SMILES representation (O’Boyle. "Towards a Universal SMILES representation - A standard method to generate canonical SMILES based on the InChI". Journal of Cheminformatics, 2012, 4, 22)
>
> Comment: (6) They author doesn't provide a code to reproduce their experiments.
>
> Response: We thank the reviewer for the thoughtful comment. We add the URL of the code in https://github.com/bmebmebme/ms2transformer.

---

> > ### Comment · Reviewer_PCDA · 2021-11-23
> > **Data spliting Issue**
> >
> > Thanks for explaining how you split the spectrum. However, I don't think an average intra-molecule cosine of 0.38 means you should randomly split the dataset. The cosine similarity between intra-molecule spectra could be low due to the different intensity distribution over peaks. However, the occurrence of peaks among those spectra are very similar and those are the key indicators for identifying molecule structure. As you said in your paper, database matching fails most of the time. Therefore, you always expect to identify some unseen molecule. Can you show the performance with the setting that split the dataset based on the molecule?
> >
> > Besides, you didn't mention the validation set. Did you just show the validation performance as the test performance?
> >
> > Alternatively, can you just show the generalization performance of your model (trained on massbank only) on the CASMI 2017 which has 243 molecules? I believe this provides a better benchmark since there are multiple participants' results.

---

### Official Review · Reviewer_g6pz · 2021-11-01

**Correctness:** 3
**Technical Novelty And Significance:** 3
**Empirical Novelty And Significance:** 3
**Recommendation:** 5
**Confidence:** 3

**Main Review:**

This paper addresses an important problem using a convincingly good model.  The design choices overall seem sensible, and the results appear to be strong.
However, I have two main critiques, and a number of minor comments as well.

First, this is a well-studied problem, but the the paper fails completely to situate the performance relative to the state of the art.  For example, we are told (p. 6) which performance measures were used to evaluate the system, but not whether these are the same performance measures employed in other studies (e.g., CAMSI).  It's important to use commonly accepted performance measures or to justify using other measures.  More importantly, the method is compared to the author's own implementations of several competing machine learning models. This is mildly interesting, but not nearly as interesting as would be a comparison to existing, state-of-the-art methods on these same benchmark datasets.  Without such comparisons, it's impossible to evaluate the practical impact of this work.

Second, the description of the peak embedding module (Section 3.1) is not clear at all.  I get that E_{m/z} is a matrix of dimension V \times d, but then I don't understand what E_i is. I think maybe it's a vector of length d, but the sentence is written as if there is a matrix of size E_i \times d. The next sentence says that V is the number of distinct m/z ratios.  I think this means that V is the number of peaks, but then it's also the number of intensities.  I don't understand why V only applies to the m/z ratios (since every m/z ratio comes paired with an intensity). The real problem is that we are told that m_i is encoded using a one-hot representation.  This is completely mysterious to me, and is related to the incorrect claim (p. 3) that m/z is discrete. The m/z ratio is a real value, measured with varying amounts of precision depending on the instrument type.  I do not understand in what sense this measurement can be considered discrete, nor how to encode it as a one-hot representation.  I am guessing that it's a vector where each entry corresponds to a mass bin, but then why aren't we told what bin size is used?  This seems like a critical parameter, and for realistic values it means that the vector will be really long (e.g., hundreds of thousands).  Given all the problems in the first 1.5 sentences of this subsection, I really can't make sense of Equation (1) nor the subsequent description of the encoder.

The claim (p. 1) that "about 90% [of] spectrum molecules is not covered by the standard database" is problematic.  No citation is given to support this claim. Plus, it surely cannot be true in general, since the coverage will depend strgonly on the type of sample being analyzed and the choice of database.

The fragmentation tree shown in Figure 1B is not described well enough.  More detail needs to be provided somewhere in the paper about how the initial graph is constructed and exactly what the edges mean,

At first I thought that Table 1 and Figure 4 seemed redundant and that the table should be eliminated. But then I noticed that the mean values marked in Figure 4 do not agree with the mean values reported in Table 1. So I think I must be misunderstanding something here, which is worrisome.

We are told on p. 8 that the model achieves its performance "in a light-cost [sic] way" using only "an additional 0.26 M parameters." What is this relative to?

The fragmentation inference task in Section 4.4 is not well described. I do not really understand what exactly the problem is here. More importantly, no baseline method is included for this task, so we do not know whether the proposed model improves on the state of the art.

The use of English is problematic throughout.  Careful editing is needed to fix grammatical errors.  Here are a few:

Abstract: "demonstrating it the great"

p. 1: "progresses have been achieved"

p. 5: "uniformed sampled"

In Figure 1A, the fragmentation is not always accomplished via a laser.



**Summary Of The Paper:**

This paper proposed to a use a deep learning model with a transformer architectur to translate between mass spectra and chemical structures of unknown metabolites.

**Summary Of The Review:**

I think this might be very good work, but the explanation of the model is problematic, and the empirical results fail to explain how the model's performance compares to the state of the art.

---

> ### Author Response · Authors · 2021-11-23
> **Response to Reviewer g6pz**
>
> We thank the reviewer for his/her valuable comments. Due to the character limits, we put the start and the end of the comments in the response.
>
> Comment: First, this is a well-studied problem ...... to evaluate the practical impact of this work.
>
> Response: Firstly, we investigated the evaluation approaches in other studies. There are several widely-used evaluation approaches: F1 score, mean/median rank, top1/3/10 accuracy, mean/median RRP. We added these approaches to further evaluate MS^2-Transformer’s performance and found that MS^2-Transformer still achieved the best performances in these metrics.
>
> Secondly, as for the comparison of the state-of-art methods, we further compared the identification performance of MS^2-Transformer with MetFrag, which represents the conventional working model for MS/MS-assisted molecule identification. We tested the performances of MetFrag by searching the molecules from PubChem and KEGG databases with a mass accuracy of 0.1 Da. Compared to MetFrag, all deep learning models showed superior performances on the metrics.
>
> Comment: Second, the description of the peak embedding ...... the encoder.
>
> Response: Every spectrum was binned in the m/z range between 1 Da and 500 Da with a bin size of 1 Da.  E_I is the representation of intensities, coming from a linear transformation of raw intensities, where every intensity is transformed from a scalar to a d-dimensional vector. And Peak Embedding Module works as follows:
>
> Step 1: construction of m/z embeddings by binned m/z ratios with a bin size of 1 Da;
>
> Step 2: construction of intensity representation by a linear transformation, where every intensity is projected from a scalar to a d-dimensional vector;
>
> Step 3: construction of peak embedding by concatenate operation and linear transformation with the above m/z embedding and intensity representation.
>
> We revised the related descriptions in the manuscript.
>
> Comment: The claim (p. 1) that "about 90% ... database.
>
> Response: We have listed the related references to support the claim of "about 90% of spectrum molecules is not covered by the standard database" (Shen, et al., Nature Communications, 2019, 10, 1516; Vinaixa, et al., TrAC Trends in Analytical Chemistry, 2016, 78, 23-35) and pointed out the databases of the above statement.
>
> Comment: The fragmentation tree shown ... edges mean,
>
> Response: The nodes in fragmentation graph represent the peaks in tandem mass spectrum, and every peak is listed with all possible molecular formulas by matching the molecular mass (so-called monoisotopic mass). The edges represent the fragmentation reactions between peaks. The calculation process of fragmentation tree includes lots of prior chemical knowledge and here is detailed description:
>
> Step 1: Initialize the fragmentation graph; enumerate all possible molecular formulas for every node by matching the monoisotopic mass;
>
> Step 2: Score the fragmentation graph; Briefly, the scoring is based on nodes and edges. For nodes, calculate the mass difference between the measured peak and the molecular formula, then compute log odds to differentiate the fragment and the background noise. And the principle behind this is that mass differences for fragments are assumed to be Normal distributed while exponentially distributed for noises. For edges, calculate the common neutral losses.
>
> Step 3: Enumerate the all the possible fragmentation trees and choose the one with highest score as the final selected fragmentation tree.
>
> We revised the related parts in manuscript.
>
> Comment: At first I thought that ... which is worrisome.
>
> Response: The statistics of #Edges of T, #Peaks of MS/MS, and SMILES length indicate the mean value with standard deviation, while the red lines in Figure 4 refer to the 95% percentile. We deleted the statistics in Table 1, which seems redundant.
>
> Comment: We are told on p. 8 ... relative to?
>
> Response: MS^2-Transformer is different from the conventional transformer in the peak embedding module and the fragmentation aggregation module, leading to an additional 0.026 M parameters.
>
> Comment: The fragmentation inference ... the state of the art.
>
> Response: The fragmentation inference task in Section 4.4 displays that MS^2-Transformer could infer the potential fragmentation reactions between peaks, indicating the strengths of chemical bonds during MS tests. However, as the theoretical research of ionization mechanism in MS is not well-studied, there is no baseline computational model for this task. To avoid further confusion on the chemical background, we deleted the related part.
>
> Comment: The use of English ... fix grammatical errors. Here are a few:
>
> Response: We revised the related descriptions in the manuscript.
>
> Comment: In Figure 1A, the fragmentation is not always accomplished via a laser.
>
> Response: We concluded that the fragmentation mainly includes collision-induced dissociation, surface-induced dissociation, and photodissociation and revised the related parts in manuscript.

---

### Official Review · Reviewer_ZGQp · 2021-11-02

**Correctness:** 4
**Technical Novelty And Significance:** 3
**Empirical Novelty And Significance:** 4
**Recommendation:** 5
**Confidence:** 4

**Main Review:**

The paper is tackle a very interesting biochemistry problem --  tandem mass spectrometry (MS/MS).  The paper provides clear description for the audience to understand the data and the problem (e.g. Fig 4 and Fig5). The three modules are interesting (e.g. encode the peak embedding and incorporate fragmentation tree structure).

Some questions:
(1) Although the three modules framework is interestingly designed, it seems to me that the contribution of machine learning novelty seems not very clear.
(2) Table 1 shows the MS/MS number of pairs seem limited, especially CASMI. Are transformer-based or deep learning models that require a lot of training data the most appropriate methods, compared with non-deep learning based methods?
(3) MS/MS is a well established technique for many years with mature data analysis pipeline (from spectrum to molecule identification), can the author provide evaluation against currently working data analysis pipeline to Table 2 and Table 3 so that the readers can understand the performance of deep learning model against conventional working model?
(4) Fig 5 a,  why the true predictions in the circles have different scores?
(5) Fig 3 SMILES-reconstruction module, why the input is smiles (shifted right). Is the smiles string the target, not the input?


**Summary Of The Paper:**


This paper provides an end-to-end model, denoted as MS2 -Transformer, using MS/MS spectrum data to specify molecule identity. The application is very important to biochemical studies.  This model consists of three modules, the peak embedding module, the fragmentation aggregation module, and the SMILES reconstruction module. The MS2 - Transformer incorporates chemical knowledge through a fragmentation tree from the MS/MS spectrum.



**Summary Of The Review:**

This paper proposes an interesting method to solve an important biochemical problem, data analysis of tandem mass spectrometry (MS/MS). The paper may has a larger scientific impact if it submit to a corresponding high impact journals, which are likely have more appropriate audience. However, if the paper is interested in machine learning conference, some more contribution of machine learning novelty and more quantification of the methods can strengthen the paper.

---

> ### Author Response · Authors · 2021-11-23
> **Response to Reviewer ZGQp**
>
> Comment: (1) Although the three modules framework is interestingly designed, it seems to me that the contribution of machine learning novelty seems not very clear.
>
> Response: We thank the reviewer for the thoughtful comment. We conclude our contribution of machine learning novelty to the insights behind Fragmentation Aggregation (FA) module with incorporation of prior chemical knowledge.
>
> The insights begin with how to learn the representation of MS/MS data. As MS/MS spectrum indicates the chemical bonds’ strengths, FA module is designed to learn the bonds’ strengths globally and locally. The global strengths of bonds are learned by information aggregation among all peaks, referred as a fully-connected graph. The local strengths of bonds are learned by the aggregation among partial peaks, which are estimated in the precursor-product relationship, referred as a locally-connected graph. The estimated precursor-product relationship is calculated with prior chemical knowledge by fragmentation tree. With the above understanding, we naturally introduce self-attention mechanism and message passing neural network for the aggregation on fully- and locally connected graph, respectively. And to our best knowledge, it is the first deep learning work on interpreting MS/MS data by introducing aggregation on bonds’ strengths.
>
> Comment: (2) Table 1 shows the MS/MS number of pairs seem limited, especially CASMI. Are transformer-based or deep learning models that require a lot of training data the most appropriate methods, compared with non-deep learning based methods?
>
> Response: We thank the reviewer for the helpful comment. We studied several biochemistry-related works with deep learning models as their sequence models and concluded that data number of tens of thousands could be enough for training. For example, McGee, et al. used 20K protein multiple sequence alignments with 10K for sequence model training (McGee, et al., "The generative capacity of probabilistic protein sequence models". Nature Communications, 2021, 12, 6302), Avsec, et al. used 29-34K genetic sequences for model training and 1.9-2K genetic sequences for testing (Avsec, et al., "Effective gene expression prediction from sequence by integrating long-range interactions". Nature Methods, 2021, 18, 1196-203), and Taujale et al. used 45K glycosyltransferases sequences in their deep learning (Taujale, et al., "Mapping the glycosyltransferase fold landscape using interpretable deep learning". Nature Communications, 2021, 12, 5656). As for the training on CASMI dataset, we fine-tuned the pre-trained models on MASSBANK dataset with CASMI dataset.
>
> Comment: (3) MS/MS is a well established technique for many years with mature data analysis pipeline (from spectrum to molecule identification), can the author provide evaluation against currently working data analysis pipeline to Table 2 and Table 3 so that the readers can understand the performance of deep learning model against conventional working model?
>
> Response: We thank the reviewer for the thoughtful comment. We added the experiments by MetFrag (Ruttkies, et al., "MetFrag relaunched: incorporating strategies beyond in silico fragmentation". Journal of Cheminformatics, 2016, 8, 3), which represents the conventional working model for MS/MS-assisted molecule identification. We tested the performances of MetFrag by searching the molecules from PubChem and KEGG databases with a mass accuracy of 0.1 Da. Compared to MetFrag, all deep learning models showed superior performances on the metrics. The results for MassBank are as follows:
>
> | Model | V\% | R\% | Top1\% | Top5\% | F1 score | Mean rank | Mean RRP |
>
> | MetFrag-PubChem | - | - | 0.10 | 0.48 | 328 | 9.96 | 0.04 |
>
> | MetFrag-KEGG | - | - | 1.14 | 1.55 | 1,405 | 9.85 | 0.02 |
>
> | RNN | 77.4 | 59.0 | 45.7 | 47.0 | 45,953 | 5.32 | 0.47 |
>
> | Transformer | 97.7 | 58.0 | 56.7 | 70.1 | 64,980 | 3.11 | 0.69 |
>
> | MS$^2$-Transformer | 97.1 | 63.2 | 61.4 | 73.7 | 68,736 | 2.77 | 0.72 |
>
>
> Comment: (4) Fig 5 a, why the true predictions in the circles have different scores?
>
> Response: We thank the reviewer for the helpful comment. Because the circled molecular graphs were predicted with different tokens after the prediction of end of sentence (EOS) token. As for the circled graphs of Fig. 5a, the differences of three predicted strings after EOS token were listed as follows:
> <EOS><EOS><EOS><EOS><EOS><EOS><EOS><EOS> for score of -2.70E-01;
> cccc21<EOS><EOS> for score of -1.42E+00;
> <EOS><EOS><EOS><EOS>cccc for score of -1.54E+00.
>
> Comment: (5) Fig 3 SMILES-reconstruction module, why the input is smiles (shifted right). Is the smiles string the target, not the input?
>
> Response: We thank the reviewer for the thoughtful comment. Because we trained the SMILES-reconstruction module with teacher forcing strategy, where the ground truth from a prior time step (shifted right) was used as input.

---

### Decision · Program_Chairs · 2022-01-20

**Decision:**

Reject

**Comment:**

Finally, all reviewers leaned towards rejection. The main concerns were missing methodological depth and questions regarding the experimental evaluation (unclear link between experimental outcomes and methodological details). The rebuttal was not perceived as being fully convincing, and finally nobody wanted to champion this paper. I think that this work has some potential, but in its present form, it does not seem to be ready for publication.